# Determination of Force Coefficients for a Submerged Rigid Breakwater under the Action of Solitary Waves

**Francesco Aristodemo** [1,*] , **Giuseppe Tripepi** [1] , **Luana Gurnari** [2] **and Pasquale Filianoti** [2]

1   Dipartimento di Ingegneria Civile, Università della Calabria, via Bucci, cubo 42B,
    87036 Arcavacata di Rende (CS), Italy; giuseppe.tripepi@unical.it
2   Dipartimento D.I.C.E.A.M., Università degli Studi Mediterranea di Reggio Calabria, via Salita Melissari,
    89124 Reggio Calabria, Italy; luana.gurnari@unirc.it (L.G.); filianoti@unirc.it (P.F.)
*   Correspondence: francesco.aristodemo@unical.it; Tel.: +39-0984-496554

**Abstract:** We present an analysis related to the evaluation of Morison and transverse force coefficients in the case of a submerged square barrier subject to the action of solitary waves. To this purpose, two-dimensional experimental research was undertaken in the wave flume of the University of Calabria, in which a rigid square barrier was provided by a discrete battery of pressure sensors to determine the horizontal and vertical hydrodynamic forces. A total set of 18 laboratory tests was carried out by varying the motion law of a piston-type paddle. Owing to the low Keulegan–Carpenter numbers of the tests, the force regime of the physical tests was defined by the dominance of the inertia loads in the horizontal direction and of the lift loads in the vertical one. Through the use of the time series of wave forces and the undisturbed kinematics, drag, horizontal inertia, lift, and vertical inertia coefficients in the Morison and transverse semi-empirical schemes were calculated using time-domain approaches, adopting the WLS1 method for the minimization of the difference between the maximum forces and the linked phase shifts by comparing laboratory and calculated wave loads. Practical equations to calculate these coefficients as a function of the wave non-linearity were introduced. The obtained results highlighted the prevalence of the horizontal forces in comparison with the vertical ones which, however, prove to be fundamental for stability purposes of the barrier. An overall good agreement between the experimental forces and those calculated by the calibrated semi-empirical schemes was found, particularly for the positive horizontal and vertical loads. The analysis of the hydrodynamic coefficients showed a decreasing trend for the drag, horizontal inertia, and lift coefficients as a function of the wave non-linearity, while the vertical inertia coefficient underlined an initial increasing trend and a successive slight decreasing trend.

**Keywords:** submerged rigid breakwater; solitary waves; physical tests; hydrodynamic forces; Morison and transverse formulas; force coefficients





## 1. Introduction

Submerged rigid or rubble-mound breakwaters have recently been adopted to defend coastal areas by the wave motion. The advantages of submerged breakwaters include the lower environmental impact and cost constructions with respect to the traditional emerged ones.

The literature regarding the interaction between solitary waves and submerged rigid breakwaters is wide. Indeed, a relevant amount of studies paid attention to different hydrodynamic aspects of the above physical phenomenon. The major part of the research was concentrated on the flow field near the roof and lee side of the barriers with a consequent formation of vortical structures [1–8]. Another aspect well-investigated referred to the evaluation of reflection, transmission, and dissipation (RTD) coefficients [1,4,7,9–14]. Less studies were instead addressed to investigate the solitary wave-induced forces at submerged barriers. With reference to only horizontal loads, Huang and Dong [2] highlighted, through the use of unsteady Navier–Stokes equations, the occurrence of higher positive

peaks and lower negative peaks, while Wu et al. [4] experimentally observed an overall linear tendency of the maximum values vs. the wave non-linearity $A/d$, with $A$ being the wave amplitude and $d$ the water depth. More recently, Tripepi et al. [15] calibrated a semi-analytical method depending on the speed drop factor, preliminarily investigated by Filianoti and Di Risio [16], to determine the horizontal loads on the basis of an experimental investigation coupled with $\delta$-LES-SPH simulations. However, the nature of the above semi-analytical approach is inertial and, for high values of amplitudes of the incoming solitary waves at the barrier, the drag force component plays a relevant role in modeling the horizontal forces (e.g., [17]). Although the calibration of four hydrodynamic coefficients is required, the use of the Morison [18] and transverse (e.g., [19]) semi-analytical schemes proves to be a good method to calculate both the horizontal and vertical hydrodynamic loads in a practical way.

In this context, 2D physical modeling was carried out in the wave channel of the University of Calabria in order to determine the horizontal and vertical hydrodynamic forces at a submerged rigid breakwater under the action of solitary waves. The importance of the present research is addressed to investigate the stability of a coastal defence structure with a low environmental impact, such as a submerged barrier. The correct understanding of the time series of the hydrodynamic forces proves to be fundamental for the stability purposes of this structure, such as the determination of the critical conditions against the sliding and the overturning which are commonly retained as the most frequent modes of failure of vertical structures, such as caisson breakwaters (see, e.g., [20,21]). In the experiments, a value of $a/d$ equal to 0.5 was adopted, where $a$ is the height of the barrier. A battery of pressure sensors installed along the external surface of the barrier was used to deduce the wave loads, and two wave gauges installed before and after the breakwater were also adopted to model the incident flow field, that is, surface elevation and flow kinematics. The force regime at the barrier was characterized by the prevalence of the inertia force component along the horizontal direction, and of the lift force component along the vertical one [22,23]. On the basis of the time variation of the hydrodynamic loads and the free stream kinematic field at the barrier, the force coefficients (drag, horizontal inertia, lift, and vertical inertia) in the Morison and transverse semi-analytical methods were calibrated by means of ordinary and weighted least square approaches [24]. The choice of $a/d = 0.5$ is due to the fact that the adopted semi-empirical equations prove to be effective when the surface elevation is not substantially deformed by the presence of the submerged barrier, as in the present case. Then, this work could be useful for engineering purposes due to the practical way of determining the hydrodynamic loads by simple semi-empirical formulas, where the calculated hydrodynamic coefficients can be determined by the knowledge of practical quantities, such as the wave amplitude and the water depth.

The current paper is organized as follows. Section 2 recalls some basic equations to model a solitary wave. The experimental tests are described in the successive Section 3. The Morison and transverse semi-empirical schemes are illustrated in Section 4. Section 5 shows the obtained laboratory results in terms of surface elevation before the barrier and related undisturbed kinematics (horizontal velocity, and horizontal and vertical acceleration), horizontal and vertical wave forces (time series and peaks), calibration of force coefficients, and application of semi-analytical formulas to reproduce the time variation of the hydrodynamic forces, paying attention to the maximum values and the related phase shifts.

## 2. The Solitary Wave

The solitary wave is given by a unique gravity wave which proves to be totally above the sea water level. Its shape can be assimilated to certain characteristics of the leading wave of a tsunami wave train, storm surges, and other types of long waves [2]. The representation of a solitary wave is easy and robust, and has been analyzed by different theoretical methods (e.g., [25–28]) and applied experimentally and numerically for different purposes.

Following the first approximation order introduced by Laitone [27] and Grimshaw [28] in the limit $d/L \rightarrow 0$ where L is the wave length, the surface displacement of a solitary wave is due to

$$\eta(x,t) = \frac{A}{cosh^2[k(x-ct)]}, \tag{1}$$

where $x$ is the direction of the wave motion, $t$ is the time, $k$ is the wave number, and $c$ is the wave celerity.

The value of $k$ is dependent on $A$ and $d$, and calculated as

$$k = \sqrt{\frac{3A}{4d^3}}. \tag{2}$$

The value of $c$ is calculated by the following relationship:

$$c = \sqrt{gd}\left(1 + \frac{A}{2d}\right) \tag{3}$$

with $g$ being the acceleration gravity.

For the successive calculation of the hydrodynamic coefficients, it is useful to determine the horizontal velocity $u$ and acceleration $a_H$, and the vertical acceleration $a_V$. These quantities are calculated, in the first approximation order, as follows [27,28]:

$$u(x,t) = \frac{\sqrt{gd}}{d}\eta(x,t) \tag{4}$$

$$a_H(x,t) = \frac{\sqrt{gd}}{d}\frac{2cksinh[k(x-ct)]}{cosh[k(x-ct)]}\eta(x,t) \tag{5}$$

$$a_V(x,t) = 2ck^2\sqrt{gd}\eta(x,t) - \frac{6ck^2sinh^2[k(x-ct)]\sqrt{gd}}{cosh^2[k(x-ct)]}\eta(x,t). \tag{6}$$

Theoretically, the value of $L$ and the wave period $T$ of a solitary wave are infinite. However, for practical calculations, an apparent wave length and period can be determined. In this context, a finite wave length $L = 2\pi/k$ and a finite wave period $T = L/c$ are defined. In this way, at a distance $x = L/2$ from the wave crest, the surface elevation is 0.74% of $A$ (e.g., [29]).

In this work, the generation of the solitary wave due to a piston-type paddle is due to the following equation, to determine its horizontal movement [30]:

$$X(t) = \frac{A}{kd}tanh\{k[ct - X(t)]\}. \tag{7}$$

## 3. Laboratory Tests

The experimental tests were undertaken in the wave channel of the Department of Civil Engineering of the University of Calabria (Italy). The adopted flume is 41 m long, of 1 m width, and 1.2 m depth, and characterized by a piston-type wavemaker, loading and unloading tanks, and a final rubble-mound breakwater to partially dissipate the waves. The wave flume is made by 15 modules, suitably joined and waterproofed, with a length of about 2.7 m. Its skeleton is made by steel, while the walls and the bottom are made of glass. Regarding the piston-type wavemaker, it is moved by a servo-controller hydraulic actuator with a maximum stroke of 0.5 m. In particular, the paddle movement is controlled indirectly by the rotation of a joint of the mechanical chain, that is connected to the paddle. The rotation angle is measured with a resistive encoder that provides a proportion analog voltage signal. This signal is processed, as well as the set-point signal, by a properly tuned Proportional Integral Derivative (PID) controller. The PID acts in order to minimize the error, that is, the difference between the set-point and the feedback signals. The output of the PID is connected to the kinematic chain through a hydro-pneumatic actuator. The

set-point signal is generated by a Data AcQuision Board (DAQ), thanks to a Digital-to-Analog Converter (DAC) (see, for more details, [31]). Note that this device is similar to the wave generation system available at the University of Louisiana at Lafayette [32,33]. The longitudinal profile of the adopted experimental set-up is highlighted in Figure 1.

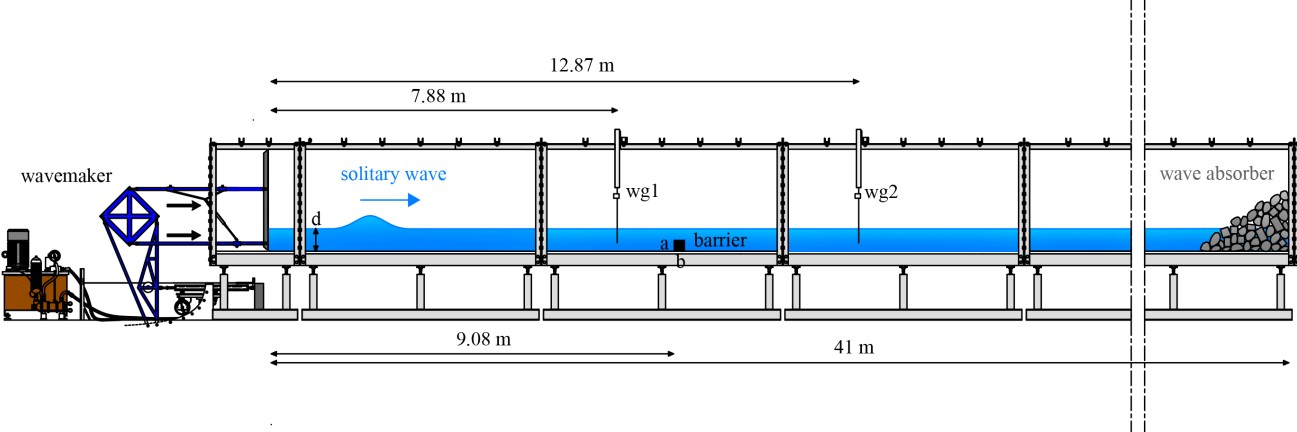

**Figure 1.** Longitudinal profile of the adopted experimental set-up.

The rigid submerged barrier was made of iron and covered by a thin layer of an electrolytic zinc plating to avoid corrosion by water. Its shape was square, with a height of $a$ = 0.127 m and length $b$ = 0.127 m, and its installation was at 9.08 m from the wavemaker, that is, more than one wave-length. The lower parts of the barrier were fixed at the bed through a special silicone. To deduce the hydrodynamic forces acting on the barrier, four pressure transducers were arranged at its beaten side, four at the roof of the structure, and four at its lee side (Figure 2a). Owing to the constructional constraints, the pressure sensors at the beaten and lee sides were transversally staggered with a mutual distance of a few cm, as performed in similar experiments on circular cylinders by Aristodemo et al. [22,34]. This trick was possible because of the 2D nature of the wave fronts of the solitary waves. A detail of the installation of the pressure transducers at the barrier from the beaten side is shown in Figure 2b.

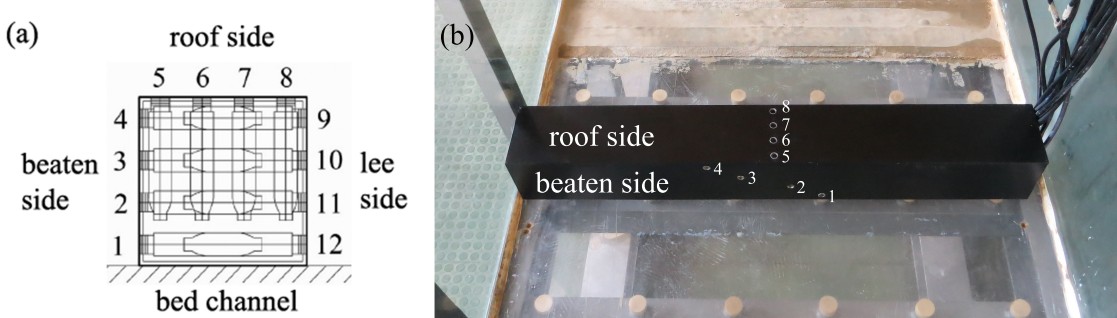

**Figure 2.** (**a**) Representative cross-section of the pressure transducers at the barrier; (**b**) Detail of the installation in the wave flume of the pressure transducers at the barrier from the beaten side.

On the basis of the geometrical disposition of the pressure sensors at the barrier and considering a piecewise linear variation of pressure along each side of the barrier [15], the horizontal and vertical hydrodynamic loads are evaluated as

$$\left\{ \begin{array}{l} F_H(t) = A_1[\Delta p_1(t) + \Delta p_2(t) + \Delta p_3(t) + \Delta p_4(t) - \Delta p_9(t) - \Delta p_{10}(t) - \Delta p_{11}(t) - \Delta p_{12}(t)] \\ F_V(t) = A_2[\Delta p_1(t) + \Delta p_{12}(t)] - A_1[\Delta p_5(t) + \Delta p_6(t) + \Delta p_7(t) + \Delta p_8(t)] \end{array} \right\}, \tag{8}$$

where $A_1$ and $A_2$ are the influence areas and equal to $a/4$ and $a/2$, respectively. It is worth noting that in the calculation of the vertical loads, a trapezoidal pressure distribution

at the base of the barrier was taken into account. This approach represents a widely adopted method to deduce the above forces for stability purposes (see, e.g., [21]). Regarding the number of the pressure gauges used in the analysis to determine the hydrodynamic forces, a very small difference was observed, about the use of 12 pressure sensors or a higher number because of the presence of long waves, such as the solitary waves which were of slow impact since the adopted barrier showed relevant submergence. The aspect related to the number of the pressure gauges was highlighted through numerical simulations in a previous work [15]. Indeed, the use of a high number of pressure gauges around the external contour of the barrier, that is, 150, showed very similar results with those obtained using 12 pressure sensors in the calculation of the hydrodynamic forces, particularly in correspondence with the peaks.

The surface elevations were measured by two resistive wave gauges (wg1 and wg2) placed respectively at a distance of 1.2 m before the rigid breakwater, and of 3.66 m after the rigid breakwater (see Figure 1). The sampling frequency $f$ of the pressure transducers and the wave gauges was set at 100 Hz.

A total number of 18 laboratory experiments was carried out. The still water level was equal to $d = 0.254$ m so that the ratio $a/d$ was 0.5. Table 1 highlights the laboratory values of $A$, $T$, $A/d$, $KC = u_m T/a$, and $Re = u_m a/\nu$, respectively, where $u_m$ is the maximum horizontal velocity and $\nu$ is the kinematic viscosity of water. The experimental range of $A/d$ was between 0.135 and 0.326, while that related to the Keulegan–Carpenter number $KC$ and Reynolds number $Re$ ranged from 5.00 to 7.19, respectively, and from $2.71 \times 10^4$ to $6.53 \times 10^4$. The size of the experimental set-up refers to small-scale laboratory tests with an approximate Froude scale of 1:40. On the basis of the generated waves for the present experiments, this leads to realistic conditions characterized by solitary waves with wave amplitudes up to 3.3 m and wave periods up to 18.8 s interacting with a submerged barrier with a height of 5.08 m and length of 5.08 m on a water depth of 10.2 m.

**Table 1.** Main characteristics of the experimental tests.

| Test Number | A (m) | T (s) | A/d | KC | Re |
|:---:|:---:|:---:|:---:|:---:|:---:|
| 1 | 0.034 | 2.98 | 0.135 | 5.00 | $2.71 \times 10^4$ |
| 2 | 0.036 | 2.91 | 0.141 | 5.10 | $2.83 \times 10^4$ |
| 3 | 0.041 | 2.69 | 0.162 | 5.42 | $3.25 \times 10^4$ |
| 4 | 0.046 | 2.52 | 0.181 | 5.69 | $3.64 \times 10^4$ |
| 5 | 0.050 | 2.39 | 0.198 | 5.90 | $3.98 \times 10^4$ |
| 6 | 0.054 | 2.30 | 0.212 | 6.07 | $4.25 \times 10^4$ |
| 7 | 0.055 | 2.29 | 0.215 | 6.10 | $4.30 \times 10^4$ |
| 8 | 0.060 | 2.17 | 0.235 | 6.33 | $4.71 \times 10^4$ |
| 9 | 0.063 | 2.10 | 0.248 | 6.47 | $4.97 \times 10^4$ |
| 10 | 0.064 | 2.08 | 0.252 | 6.51 | $5.06 \times 10^4$ |
| 11 | 0.068 | 2.01 | 0.267 | 6.66 | $5.36 \times 10^4$ |
| 12 | 0.070 | 1.96 | 0.277 | 6.76 | $5.55 \times 10^4$ |
| 13 | 0.071 | 1.95 | 0.280 | 6.79 | $5.61 \times 10^4$ |
| 14 | 0.075 | 1.89 | 0.295 | 6.93 | $5.92 \times 10^4$ |
| 15 | 0.077 | 1.86 | 0.302 | 6.98 | $6.04 \times 10^4$ |
| 16 | 0.080 | 1.81 | 0.315 | 7.10 | $6.31 \times 10^4$ |
| 17 | 0.081 | 1.80 | 0.319 | 7.14 | $6.40 \times 10^4$ |
| 18 | 0.083 | 1.78 | 0.326 | 7.19 | $6.53 \times 10^4$ |

## 4. Morison and Transverse Semi-Empirical Schemes

A widespread method to easily evaluate the horizontal forces on a structure in a marine environment is given by the Morison scheme [18]. This formulation is usually applied for slender bodies as in the present case, that is, when $KC$ is generally greater than 4 and the vortex shedding regime around the body arises [35]. Following Morison et al. [18], the horizontal force, $F_H$, is given by the superimposition of a drag component, $F_D$, and a horizontal inertia one, $F_{HI}$. The former component occurs for the resistance of a solid body

to the incoming external motion, while the latter one is generated by the presence of an external horizontal acceleration field in correspondence with the solid body. In its general form, the total in-line force, $F_H$, is then equal to [18]

$$F_H(t) = F_D(t) + F_{HI}(t) = \frac{1}{2}\rho C_D A_c u(t)|u(t)| + \rho C_{MH} V a_H(t), \tag{9}$$

where $C_D$ is the drag coefficient and $C_{MH}$ is the horizontal inertia coefficient, while $u$ and $a_H$ are the undisturbed values of horizontal velocity and acceleration, respectively. For a square barrier, $A_c$ (cross-sectional area) = $a$ and $V$ (volume) = $a^2$.

Regarding the vertical load, the transverse force, $F_V$, is given by the sum of the lift component, $F_L$, and the vertical inertia one, $F_{VI}$. The former component is due to the rise of velocity field across a structure induced by the blocking of the flow, while the latter one is related to the occurrence of an undisturbed vertical acceleration field at the body. Similarly to the Morison equation, this load is calculated to the following relationship (e.g., [17])

$$F_V(t) = F_L(t) + F_{VI}(t) = \frac{1}{2}\rho C_L A_c u^2(t) + \rho C_{MV} V a_V(t), \tag{10}$$

with $C_L$ being the lift coefficient and $C_{MV}$ being the vertical inertia coefficient, while $a_V$ is the free stream vertical acceleration.

The free stream horizontal velocity and horizontal and vertical acceleration in the Morison and transverse formulas were evaluated by Equations (4)–(6), respectively, from the experimental values of $\eta$ recorded by the wave gauge installed before the barrier.

## 5. Experimental Results

### 5.1. Incident Flow Field

In this Section, the flow field induced by the propagation of the solitary wave along the channel is analyzed. In particular, the surface elevation and the kinematic field (horizontal velocity, horizontal acceleration, and vertical acceleration) before the barrier and related to wg1 are taken into account. In this context, the suitable evaluation of the solitary wave loads at the barrier through the use of semi-empirical equations, that is, Morison and transverse, is dependent on the above-mentioned flow variables. Considering tests no. 8 and 18 characterized by a lower amplitude and a wider period and by a higher amplitude and a shorter period, respectively, Figure 3 describes the time variation of the experimental surface elevation $\eta$ recorded by wg1 and its comparison with the analytical solution expressed by Equation (1). The reference time $t_{wg1} = 0$ refers to the passage of the solitary wave crest at wg1. For both cases, an overall good agreement between the experimental $\eta$ and the reference analytical solution can be noticed. This is particularly evident for the peak values and for the increasing part of the solitary wave. Regarding the decreasing part of the solitary waves, a certain discrepancy between laboratory tests and analytical formula is linked to the instantaneous truncation in the time law of the piston-type paddle. This drawback leads to the occurrence of spurious trailing waves represented by an irregular wave packet of small entity (e.g., [30]). Owing to the specific placement of wg1, that is, 1.2 m before the barrier, it is also possible to observe that the first trailing wave is incidentally superimposed to the reflected wave component due to the wave–structure interaction [14]. Although these observed surface fluctuations induced spurious wave forces, the current analysis was not influenced, since the attention was focused on the forces' peaks, which occurred before these unwanted oscillations.

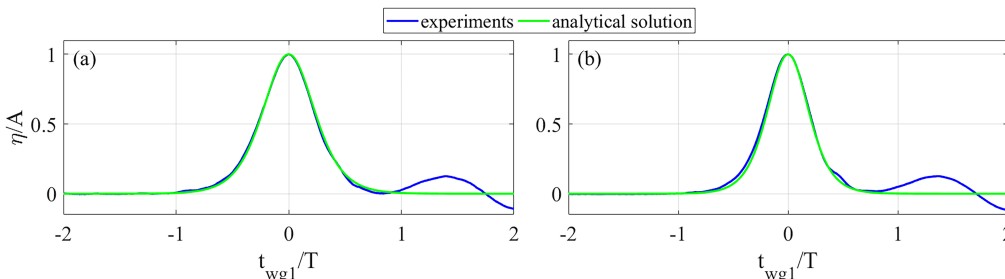

**Figure 3.** Comparison between experimental and analytical time series of surface elevation $\eta$ at wg1. (**a**) Test no. 8; (**b**) test no. 18.

Regarding the free stream kinematic field, Figure 4 shows, for the same experimental tests adopted for the surface elevation, that is, tests no. 8 and 18, the comparison between experimental and analytical time series of horizontal velocity $u$, horizontal acceleration $a_H$, and vertical acceleration $a_V$ at wg1. The experimental values of $u$, $a_H$, and $a_V$ were deduced from the recorded $\eta$ by applying Equations (4)–(6). As for the comparison of $\eta$, the agreement between experiments and analytical formulas is satisfactory, unless their final part for the reasons was previously highlighted. Concerning the horizontal velocity (see Figure 4a,b), its shape is the same as that observed for $\eta$. As shown in Figure 4c,d, the horizontal acceleration instead exhibits a positive peak before the occurrence of the wave crest and a negative peak of the same magnitude after it. Lastly, the vertical acceleration highlights two positive peaks and one higher negative peak between them, corresponding to the maximum surface elevation (see Figure 4e,f). The importance of the above kinematic components arises from its shape similarity if compared to the wave load components modeled by the Morison and transverse semi-empirical schemes. Indeed, the horizontal velocity is proportional to the drag and lift force components, the horizontal inertia load follows the shape of the horizontal acceleration, and the vertical acceleration is similar to the vertical inertia force.

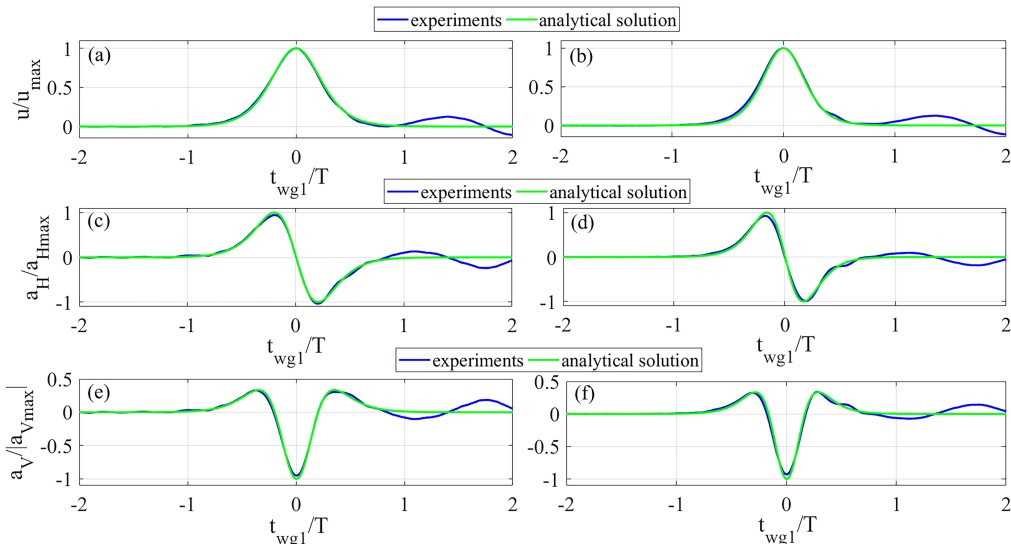

**Figure 4.** Comparison between experimental and analytical time series of kinematic field at wg1. (**a**) $u$ (test no. 8); (**b**) $u$ (test no. 18); (**c**) $a_H$ (test no. 8); (**d**) $a_H$ (test no. 18); (**e**) $a_V$ (test no. 8); (**f**) $a_V$ (test no. 18).

### 5.2. Wave Loads

The time variation and the peaks of horizontal $F_H$ and vertical $F_V$ forces at a submerged square barrier under the action of solitary waves will be examined in this Section.

For the reference tests no. 8 and 18, the time series of the hydrodynamic loads will be successively illustrated. In this analysis, the reference time $t = 0$ is linked to the appearing of the solitary wave crest at the middle of the barrier. Due to the lack of a laboratory wave gauge at this specific location because of the impossibility to install it because of the bulky presence of the barrier, a virtual measurement of surface elevation and kinematic field was determined from the recorded surface elevations at wg1 and wg2. Firstly, the wave celerity $c$ was deduced as the ratio between the distance between the two wave gauges and the time interval related to the occurrence of the maximum surface elevations at wg1 and wg2. To support this analysis, the experimental wave celerity was compared with the theoretical one calculated by Equation (3) for all experimental tests, as shown in Figure 5. It is possible to observe that the laboratory and theoretical values are quite close, with a small underestimation given by the experimental ones. The mean relative difference is equal to 2.7%. Then, the surface elevation and the kinematic field were forward-shifted by the ratio between the distance between the location of wg1 and the middle of the breakwater and the experimental celerity.

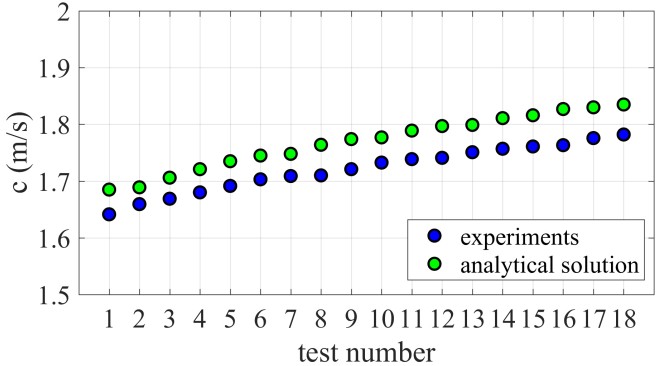

**Figure 5.** Comparison between theoretical and experimental wave celerity for all experimental tests.

The time variation of the hydrodynamic forces is shown in Figure 6 for the two mentioned tests. It can be noticed that the horizontal force is higher than the vertical one for both wave conditions. Furthermore, the horizontal force shows a greater positive peak and lower negative one, while the vertical one exhibits just one positive force peak. The occurrence of higher positive peaks and lower negative ones in the time series of the horizontal force is observed for all experimental tests, and also by other similar literature studies [2,4]. This feature can be explained by the fact that the wave forcing is characterized by a solitary wave, which has just a crest and then the barrier is substantially subject to the action of a forward motion. In the case of regular and irregular waves where crests and troughs are present, the difference between higher positive peaks and lower negative ones is generally reduced (e.g., [22]). It can also be observed that the in-line force is back-shifted with respect to the reference time $t = 0$, because of the relevant contribution of the horizontal inertia force proportional to $a_H$ if compared to the drag one. This effect is more pronounced for test no. 8 with respect to test no. 18 because of its low $KC$ number [22,35]. Regarding the vertical loads, the peak appears very close to the reference time $t = 0$ for both the test cases and, in particular, the maximum values of $F_V$ are slightly back-shifted. The overall shape of the vertical forces resembles that of the lift component which is proportional to $u$ and then to $\eta$. However, a significant broadening over the time of $F_V$ with respect to $u$ can be noticed.

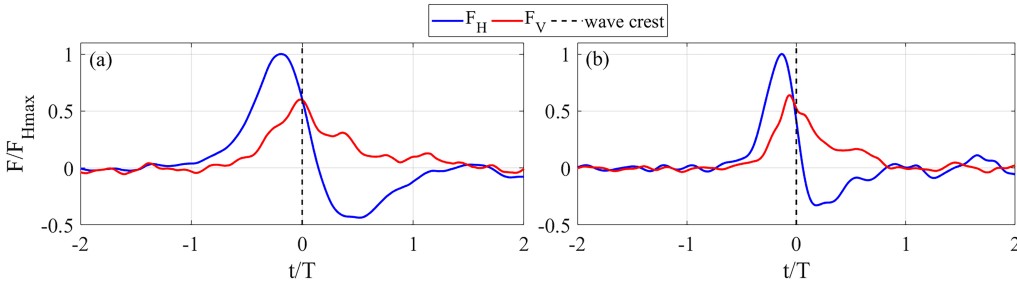

**Figure 6.** Time variation of horizontal $F_H$ and vertical $F_V$ wave forces vs. $A/d$. (**a**) Test no. 8; (**b**) test no. 18.

Considering all 18 laboratory tests, Figure 7 describes the maximum positive and negative wave forces as a function of the wave non-linearity $A/d$. The peak values are here respectively normalized with respect to the maximum peak of the horizontal force, $F_{Hmax}*$, and the maximum peak of the vertical force, $F_{Vmax}*$, occurring in the present experimental dataset. Specifically, Figure 7a refers to the peaks of positive, $F_{Hmax,p}$, and negative, $F_{Hmax,n}$, horizontal loads. The values of $F_{Hmax,p}$ exhibit an overall linear increasing trend, as also observed by Wu et al. [4]. The values of $F_{Hmax,n}$ show an initial increasing tendency, followed by a substantial stabilization for $A/d > 0.25$. Paying attention to the positive peaks of the vertical force, $F_{Vmax}$ (see Figure 7b), their trend is similar to that observed for $F_{Hmax,p}$. In order to highlight the importance of the vertical loads for stability purposes, the weight of the vertical force with respect to the horizontal one in terms of peak values is plotted in Figure 7c. It can be noticed that the weight of the vertical load oscillates between 45% for low $A/d$ and 64% for high $A/d$.

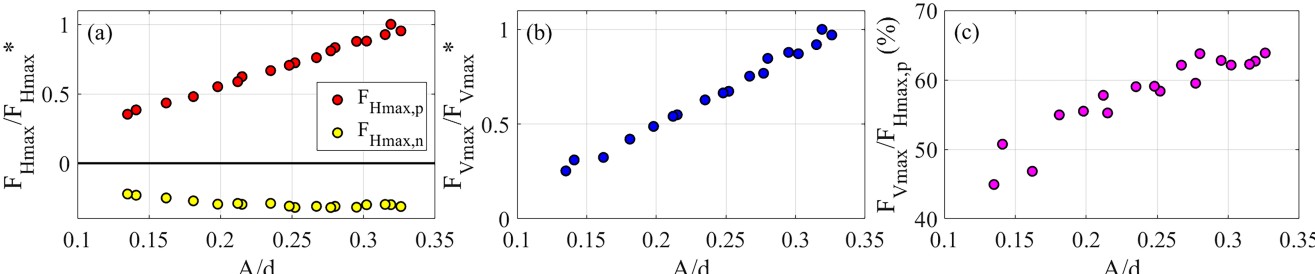

**Figure 7.** Maximum positive and negative wave forces vs. $A/d$. (**a**) Horizontal force; (**b**) vertical force; (**c**) weight of the vertical force with respect to the horizontal one.

### 5.3. Calibration of Morison and Transverse Formulas

The suitable use of the Morison and transverse semi-empirical schemes to respectively evaluate the horizontal and vertical loads induced by solitary waves at the submerged rigid breakwater requires the calibration of the hydrodynamic coefficients. Owing to the simple structure of these equations in which the kinematic field refers to undisturbed conditions, the hydrodynamic coefficients are representative parameters of the complex flow field given by this wave–structure interaction process. The determination of $C_D$, $C_{MH}$, $C_L$, and $C_{MV}$ is dependent on the experimental values of the free stream kinematic field, that is, $u$, $a_H$, and $a_V$, and of the horizontal and vertical hydrodynamic forces. The analysis was performed within the wave period, and ordinary and weighted least square methods were applied to deduce the hydrodynamic coefficients in order to minimize the differences between measured and calculated forces (see, for more details, [24]). The application of the weighted least square method is due to the fact of furnishing more emphasis to the force peaks, which are useful for stability aims of the involved marine structure.

Considering the Morison formula, the horizontal hydrodynamic coefficients $C_D$ and $C_{MH}$ are given by

$$
\left\{
\begin{array}{l}
C_D = \dfrac{\sum_{i=1}^{N} F_H^{2j+1} u|u| \sum_{i=1}^{N} F_H^{2j} a_H^2 - \sum_{i=1}^{N} F_H^{2j+1} a_H \sum_{i=1}^{N} F_{He}^{2j} u|u|a_H}{K_D\left[\sum_{i=1}^{N} F_H^{2j} u^4 \sum_{i=1}^{N} F_H^{2j} a_H - \left(\sum_{i=1}^{N} F_H^{2j} u|u|a_H\right)^2\right]} \\[3ex]
C_{MH} = \dfrac{\sum_{i=1}^{N} F_H^{2j+1} a_H \sum_{i=1}^{N} F_H^{2j} u^4 - \sum_{i=1}^{N} F_H^{2j+1} u|u| \sum_{i=1}^{N} F_{He}^{2j} a_H u|u|}{K_{MH}\left[\sum_{i=1}^{N} F_H^{2j} a_H^2 \sum_{i=1}^{N} F_H^{2j} u^4 - \left(\sum_{i=1}^{N} F_{He}^{2j} a_H u|u|\right)^2\right]}
\end{array}
\right\},
\tag{11}
$$

where $K_D = \frac{1}{2}\rho a$ and $K_{MH} = \rho a^2$. The value of $N$ represents the total number of force and kinematic values within the wave period, while $j$ is a positive index.

Taking into account the transverse equation, the vertical hydrodynamic coefficients $C_L$ and $C_{MV}$ are due to

$$
\left\{
\begin{array}{l}
C_L = \dfrac{\sum_{i=1}^{N} F_V^{2j+1} u^2 \sum_{i=1}^{N} F_V^{2j} a_V^2 - \sum_{i=1}^{N} F_V^{2j+1} a_V \sum_{i=1}^{N} F_V^{2j} u^2 a_V}{K_L\left[\sum_{i=1}^{N} F_V^{2j} u^4 \sum_{i=1}^{N} F_V^{2j} a_V - \left(\sum_{i=1}^{N} F_V^{2j} u^2 a_V\right)^2\right]} \\[3ex]
C_{MV} = \dfrac{\sum_{i=1}^{N} F_V^{2j+1} a_V \sum_{i=1}^{N} F_V^{2j} u^4 - \sum_{i=1}^{N} F_V^{2j+1} u^2 \sum_{i=1}^{N} F_V^{2j} u^2 a_V}{K_{MV}\left[\sum_{i=1}^{N} F_V^{2j} a_V^2 \sum_{i=1}^{N} F_V^{2j} u^4 - \left(\sum_{i=1}^{N} F_V^{2j} u^2 a_V\right)^2\right]}
\end{array}
\right\},
\tag{12}
$$

where $K_L = K_D$ and $K_{MV} = K_{MH}$. The case with $j = 0$ refers to the use of the ordinary least square approach.

The capabilities of the above time-domain methods to determine the various hydrodynamic coefficients were tested through the calculation of the Percentage Error (PE) obtained by the comparison between experimental and calculated hydrodynamic loads, as follows:

$$
PE = \left|\frac{F_e - F_c}{F_e}\right| * 100,
\tag{13}
$$

where $F_e$ and $F_c$ represent the generic experimental and semi-empirical hydrodynamic forces, respectively.

In the present analysis, the attention was turned to the maximum forces and, in particular, to the positive and negative peaks of the horizontal load and to the positive peak of the vertical load, and the linked time shifts, $\phi$, calculated as

$$
\phi = \frac{2\pi t_{max}}{T},
\tag{14}
$$

with $t_{max}$ being the appearing time of the peaks of horizontal or vertical loads within the wave period.

The results obtained by the calculation of PE involving the ordinary least square method, namely OLS ($j = 0$) and the weighted least square methods, namely WLS1 ($j = 1$), WLS2 ($j = 2$), WLS3 ($j = 3$), WLS4 ($j = 4$), WLS5 ($j = 5$), and WLS6 ($j = 6$), are plotted in Figure 8. The use of the weighted least square approaches up to $j = 6$ is due to the possibility to well-catch the peaks of the wave loads. Note that the values of PE are averaged considering all laboratory tests. Regarding the obtained results on PE, it is possible to observe that the values of PE related to $F_{Hmax,p}$ decrease when $j$ increases with 2.8% < PE < 9.2%. By contrast, the values of PE linked to $F_{Hmax,n}$ show a strong increasing trend proportional to $j$, starting from PE about or equal to 29% for $j = 0$ to a PE about or equal to 111% for $j = 6$. In the case of $F_{Vmax}$, it can be noticed that the maximum PE refers to the OLS approach with a value equal 12.9%, while the minimum PE of 4.4% is related to WLS1. Paying attention to the values of $\phi$, a decreasing trend for PE linked to $F_{Hmax,p}$, as in the case of the peak values, can be observed with values of PE oscillating between 1.5% (for $j = 6$) and 9% (for $j = 0$). For $\phi F_{Hmax,n}$, the rise of $j$ leads to a corresponding increase of PE with values ranging from 10.1% to 13.5%. Considering $\phi F_{Vmax}$, the trend of PE tends

to increase from OLS to WLS2, followed by a stabilization of the values of PE. In this last case, 2.6% < PE < 3.9%. In order to choose the best calibration method to be adopted for the force coefficients in the Morison and transverse semi-empirical schemes for the suitable reproduction of the laboratory forces, a mean value of PE among the above six PE related to the force peaks and the related phase shift was determined. Comparing all approaches, WLS1, that is, the weighted least square method with $j = 1$, gives the lowest mean value of PE and, specifically, is equal to 11.7%. As a result, this specific method will be taken into account for the successive analyses related to the evaluation of the force coefficients and the application of the Morison and transverse formulas.

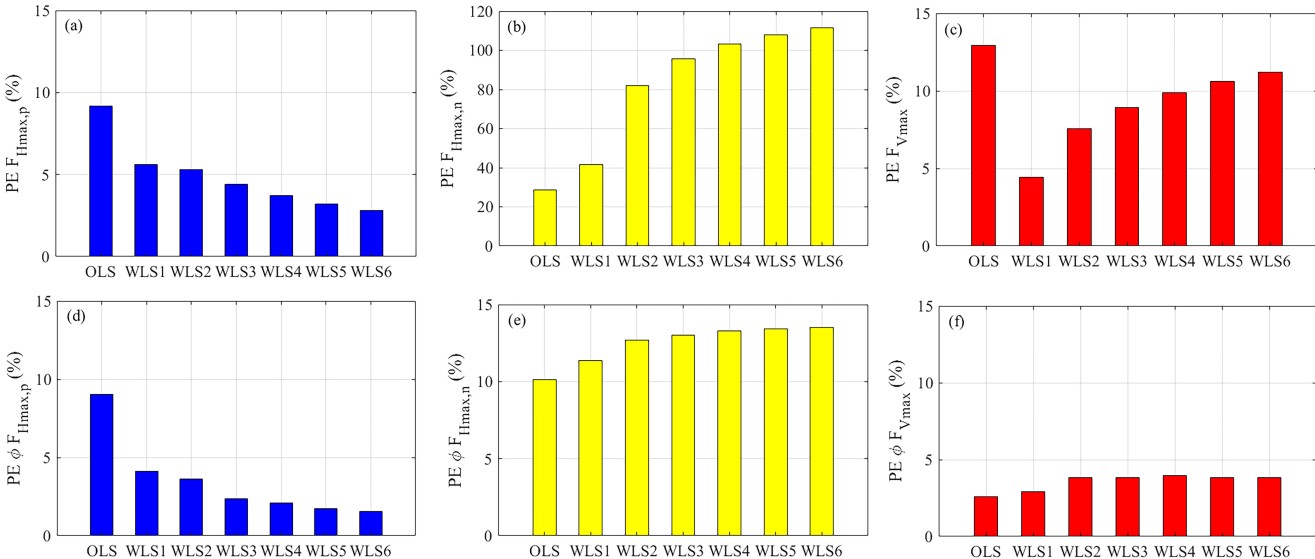

**Figure 8.** Experimental PE through OLS, WLS1, WLS2, WLS3, WLS4, WLS5, and WLS6 methods. (**a**) Maximum positive horizontal force, $F_{Hmax,p}$; (**b**) maximum negative horizontal force, $F_{Hmax,n}$; (**c**) maximum vertical force, $F_{Vmax}$; (**d**) phase shift, $\phi$, related to $F_{Hmax,p}$; (**e**) phase shift, $\phi$, related to $F_{Hmax,n}$; (**f**) phase shift, $\phi$, related to $F_{Vmax}$.

The horizontal and vertical hydrodynamic loads obtained by the Morison and transverse semi-empirical schemes through the calibration of the force coefficients using the WLS1 method are compared with the laboratory results in terms of force peaks and phase shifts linked to the peak values. In particular, Figure 9 shows the comparison between experimental and calculated maximum positive and negative wave forces as a function of the wave non-linearity. Positive and negative horizontal loads, and positive vertical loads are considered. On the basis of that obtained by PE analysis, the agreement between laboratory forces and those determined by the use of semi-empirical approaches is generally good. This is evident for the positive peaks of the horizontal, $F_{Hmax,p}$, and vertical, $F_{Vmax}$, loads. In these cases, the trend given by the peaks increases linearly vs. $A/d$, as with the experimental data. The application of WLS1 approach gives a slight overestimation of both experimental values of the positive peaks, and it can be viewed as conservative from the viewpoint of the stability of the submerged barrier. For what concerns the negative peaks of the horizontal force, $F_{Hmax,n}$, the comparison between experiments and the Morison model is fairly good until $A/d = 0.22$, with a successive overall overestimation given by the semi-empirical scheme. However, this discrepancy has no significant impact for the stability purposes of the involved structure, because of the low magnitude of these loads.

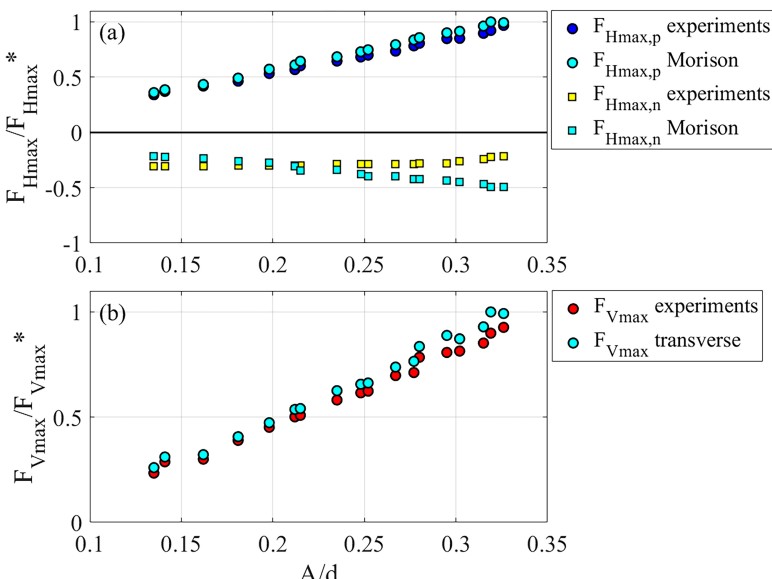

**Figure 9.** Comparison between experimental and calculated maximum positive and negative wave forces vs. $A/d$. (**a**) Horizontal force; (**b**) vertical force.

An additional comparison related to the performances of the Morison and transverse models using the hydrodynamic coefficients obtained by WLS1 is performed. In particular, it refers to the comparison between experimental and calculated phase shifts related to maximum positive and negative wave forces as a function of wave non-linearity, as illustrated in Figure 10. The different phase shifts are determined within the phase related to the wave period $(0, 2\pi)$. The plots are here shown in the order of occurrence of the time linked to the wave peaks, namely, $F_{Hmax,p}$, $F_{Vmax}$, and $F_{Hmax,n}$. Regarding $\phi$ $F_{Hmax,p}$, it is possible to notice that the values related to the Morison scheme tend to slightly forward-shift the positive horizontal peaks if compared to the experimental ones. Moreover, they occur before the half-period, that is, for $\phi < \pi$. Concerning $\phi$ $F_{Vmax}$, the transverse scheme leads to a small backward shift of the experimental vertical force peaks for $A/d < 0.16$ and a successive small forward shift for $A/d > 0.16$. This phase shift appears across the half-period or, in other words, is very close to the wave crest at the middle of the breakwater roof. Regarding $\phi$ $F_{Hmax,n}$, the Morison equation shows a backward shift of the negative peak of the horizontal load, occurring between $\pi$ and $3\pi/2$ within the wave period, in comparison with the laboratory tests. This difference is particularly evident for the central values of the studied range of $A/d$. As shown in the previous Figure 9, the result on this force peak does not affect the good results obtained for the horizontal and vertical positive peaks, which are usually adopted for the verifications of the stability of the barrier.

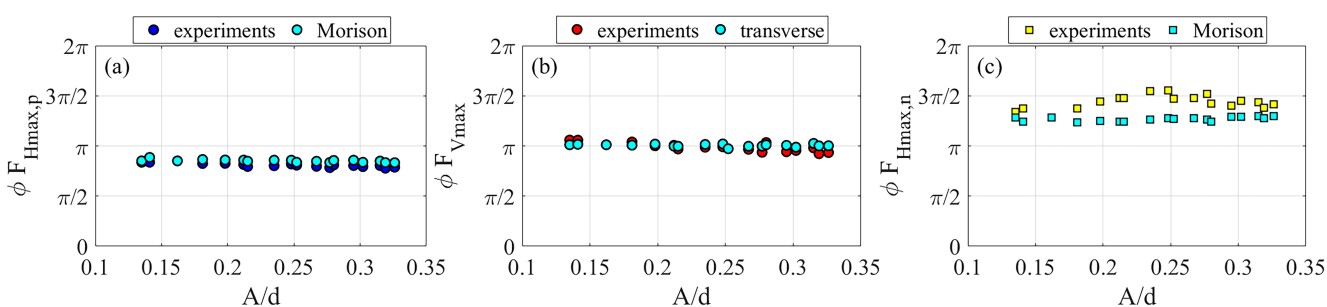

**Figure 10.** Comparison between experimental and calculated phase shifts related to maximum positive and negative wave forces vs. $A/d$. (**a**) Phase shift, $\phi$, related to $F_{Hmax,p}$; (**b**) phase shift, $\phi$, related to $F_{Vmax}$; (**c**) phase shift, $\phi$, related to $F_{Hmax,n}$.

Figure 11 shows the features of the hydrodynamic coefficients $C_D$, $C_{MH}$, $C_L$, and $C_{MV}$ obtained by the WLS1 approach as a function of the wave non-linearity. In order to assess the uncertainty in the evaluation of the hydrodynamic coefficients, the 95% prediction intervals are also plotted. The values of $C_D$ show a decreasing trend vs. $A/d$, starting from values of an order of 3 for low $A/d$ to values greater than 1 for high $A/d$. A restricted range of variability was instead observed by the results on the drag coefficients obtained by Tripepi et al. [23] in the case of solitary wave loads at a bottom-mounted cylinder with a diameter equal to the present height and length of the barrier and a different range of $A/d$. Similarly to $C_D$, the values of $C_{MH}$ decrease when $A/d$ increases. Specifically, $2.1 < C_M < 2.6$. These coefficients prove to be lower than that calculated by Tripepi et al. [23]. Paying attention to $C_L$, also in this case, the trend vs. $A/d$ is decreasing. The lift coefficients range from 4.4 for small $A/d$, to 3.3 for high $A/d$. As in the case of $C_{MH}$, the values of $C_L$ are generally smaller than those observed by Tripepi et al. [23]. Regarding the last hydrodynamic coefficient, that is, $C_{MV}$, its behaviour vs. $A/d$ is different that the other coefficients. Indeed, a rise for low $A/d$ is noticed until about $A/d = 0.2$, and a successive slight decrease when $A/d > 0.2$. The values of $C_{MV}$ oscillate between about 0.6 and 1.2. The magnitude of these coefficients is different if compared to the experiments by Tripepi et al. [23], but it proves to be comparable with the results obtained by Aristodemo et al. [36] for a solitary wave interacting with a horizontal cylinder close to the bed. Regarding the uncertainty in the assessment of the hydrodynamic coefficients, the highest is linked to the calculation of $C_{MV}$, although the weight of the vertical inertia component, $F_{VI}$, for calculating the total vertical force, $F_V$, is small if compared to the lift one, $F_L$, as successively analyzed in Section 5.4.

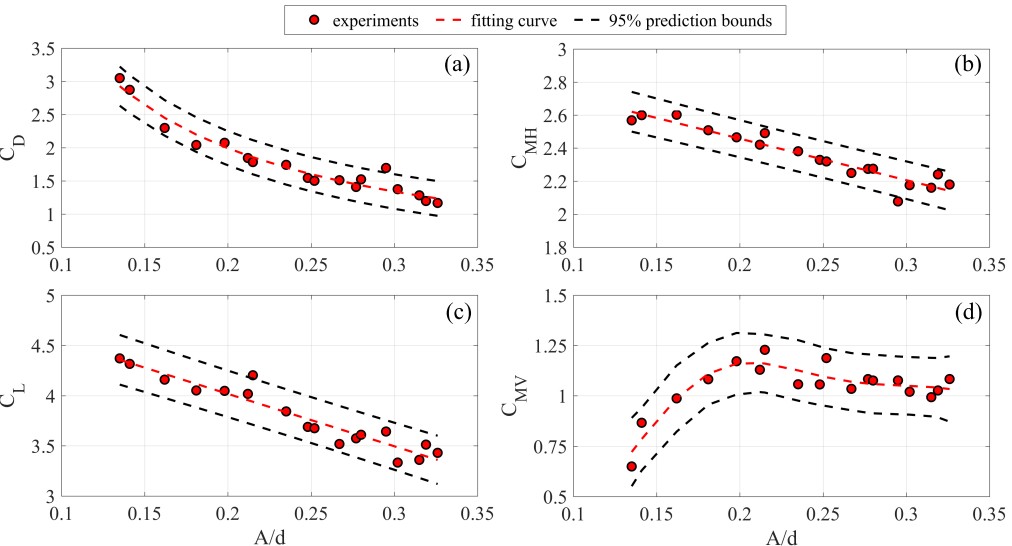

**Figure 11.** Hydrodynamic coefficients vs. $A/d$. (**a**) $C_D$; (**b**) $C_{MH}$; (**c**) $C_L$; (**d**) $C_{MV}$.

For engineering aims, empirical equations were deduced to determine, in a practical way, the hydrodynamic coefficients on the basis of the wave non-linearity. With reference to Figure 11, the fitting formulas related to $C_D$, $C_{MH}$, $C_L$, and $C_{MV}$ are highlighted through dashed red curves. The trend related to $C_D$ was well-modeled by a power law

$$C_D = 0.41 \left(\frac{A}{d}\right)^{-0.98},$$ (15)

being $R^2$ (correlation coefficient) = 0.953.

Linear laws were instead adopted to fit the experimental values of $C_{MH}$ and $C_L$

$$C_{MH} = -2.51\frac{A}{d} + 2.96$$ (16)

$$C_L = -5.22\frac{A}{d} + 5.06, \tag{17}$$

where $R^2$ is equal to 0.906 and 0.908, respectively.

A more complex law with respect to other coefficients was used to model the particular behaviour of $C_{MV}$ vs. $A/d$. Specifically, a second-order Gaussian function was adopted as follows:

$$C_{MV} = 0.65\exp\left[-\left(\frac{\frac{A}{d} - 0.18}{0.08}\right)^2\right] + 1.02\exp\left[-\left(\frac{\frac{A}{d} - 0.32}{0.16}\right)^2\right], \tag{18}$$

where $R^2 = 0.847$.

### 5.4. Application of Morison and Transverse Formulas

In this Section, the Morison and transverse formulas (see Equations (9) and (10)) were applied in order to show the contribution of each force component to model the experimental horizontal and vertical loads, respectively. The calculated forces are determined through the hydrodynamic coefficients deduced from the WLS1 approach. Figure 12 describes the experimental and calculated time series of horizontal and vertical hydrodynamic forces for tests no. 8 and 18. The Morison and transverse force components are mentioned with the subscripts M and T, respectively. In particular, Figures 12a and b show the comparison between experimental $F_H$ and Morison $F_{DM}$, $F_{HIM}$, and $F_{HM}$ for tests no. 8 and 18. It can be observed that the maximum experimental horizontal force is well-modeled by the Morison scheme with a small overestimation and forward shift. Regarding the negative peak of the horizontal load, the Morison equation gives a significant overestimation, particularly for test no. 18. As previously stressed, this peak is fairly lower than the positive one, and then is not involved for stability purposes. Concerning the horizontal force components, the inertia load is greater than the drag one. As a consequence, the peak of the laboratory horizontal force is very close and quite in phase to that related to the inertia one. Indeed, the shape of the experimental horizontal force resembles that linked to the horizontal acceleration (see Figure 4c,d), which is proportional to the horizontal inertia component. The effect of the drag force, proportional to the horizontal velocity (see Figure 4a,b), is to forward-shift the calculated horizontal load. Figure 12c,d illustrates the comparison between experimental $F_V$ and transverse $F_{LT}$, $F_{VIT}$ and $F_{VT}$ for tests no. 8 and 18. Except for a small phase shift and overestimation, the transverse scheme well-predicts the maximum value of the observed vertical force. It is possible to observe that the experimental vertical load shows a large broadening, which is not well-caught by the transverse model, particularly for test no. 18. However, this broadening effect in the transverse equation is modeled by the contribution of the vertical inertia component, which is proportional to the vertical acceleration (see Figure 4e,f). The peak of the laboratory vertical force proves to be in phase with the lift component, which is proportional to the horizontal velocity. However, the peak of the lift load is larger than the experimental vertical one and, to well-model this load, the contribution of the vertical inertia component is fundamental, since it acts to reduce the peak of the transverse model through its negative part. This effect is particularly emphasized for test no. 18.

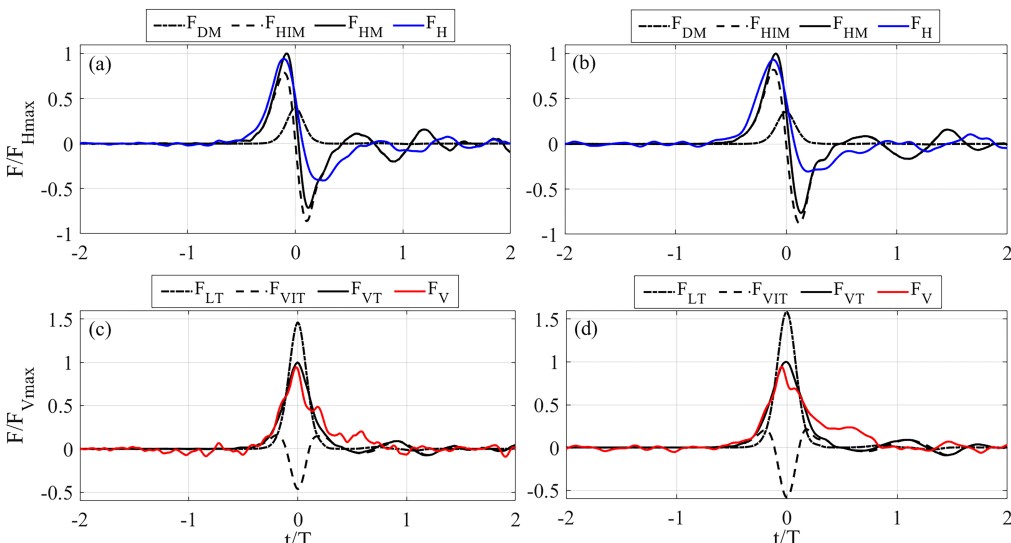

**Figure 12.** Experimental and calculated time series of hydrodynamic forces. (**a**) Comparison between experimental $F_H$ and Morison $F_{DM}$, $F_{HIM}$ and $F_{HM}$ (test no. 8); (**b**) comparison between experimental $F_H$ and Morison $F_{DM}$, $F_{HIM}$ and $F_{HM}$ (test no. 18); (**c**) comparison between experimental $F_V$ and transverse $F_{LT}$, $F_{VIT}$ and $F_{VT}$ (test no. 8); (**d**) comparison between experimental $F_V$ and transverse $F_{LT}$, $F_{VIT}$ and $F_{VT}$ (test no. 18).

As shown in Figure 13, the various force components adopted to model the Morison and transverse semi-empirical approaches, namely $F_D$, $F_{HI}$, $F_L$ and $F_{VI}$, were analyzed in terms of positive and negative peaks. In order to highlight their specific contribution, the load components are weighted with respect to the corresponding maximum horizontal and vertical forces and plotted as a function of the wave non-linearity for all 18 experimental tests. Paying attention to the horizontal force (see Figure 13a), the positive inertia components generally present with a larger weight if compared to the drag and negative inertia ones because of the low *KC* numbers of the laboratory investigation. Specifically, their weight proves to be ranging between 76% and 82% of the horizontal force and quite constant for the considered range of $A/d$. The weight of the drag component oscillates from 35% to 44% of the horizontal load, while that related to the negative inertia component shows a weight slightly greater than the positive inertia one, due to the non-perfect symmetry of the horizontal acceleration deduced from the experimental surface elevation. Regarding the vertical force contributions (see Figure 13b), it is possible to notice that the lift component shows a small increasing trend vs. $A/d$, and gives the greatest contribution in reproducing the transverse load with respect to the vertical inertia components. Moreover, the ratio between the maximum lift force and the maximum vertical one is about 1.5. The weight related to the positive inertia component is fairly small and constant vs. $A/d$. Indeed, it oscillates between 13% and 22% of the vertical load. A more significant contribution is given by the negative inertia component with peak values ranging from 32% to 59% of the vertical force. Its trend increases proportionally to the wave non-linearity.

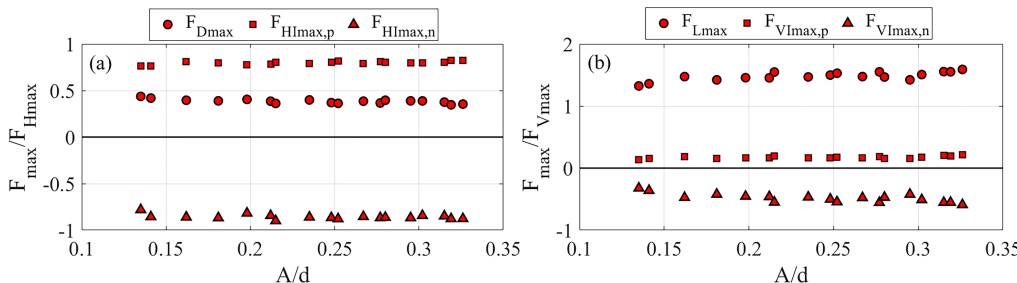

**Figure 13.** (**a**) Maximum Morison force components vs. A/d: positive drag $F_{Dmax}$, positive horizontal inertia $F_{HImax,p}$ and negative horizontal inertia $F_{HImax,n}$; (**b**) maximum transverse force components vs. A/d: positive lift $F_{Lmax}$, positive vertical inertia $F_{VImax,p}$ and negative vertical inertia $F_{VImax,n}$.

## 6. Conclusions

A 2D laboratory investigation aimed at calibrating the Morison and transverse semi-empirical models to determine, in a practical way, the horizontal and vertical hydrodynamic forces acting on a submerged rigid breakwater under the action of solitary waves has been carried out. A total number of 18 laboratory tests has been undertaken in the wave flume of the University of Calabria, where 12 pressure sensors were mounted along the external contour of a rigid square barrier to deduce the wave loads.

After a preliminary validation of incoming flow field, that is, surface elevation and kinematic field, the analysis of the force regime has underlined the prevalence of the horizontal loads in comparison with the vertical ones which, however, prove to be relevant for stability purposes. Indeed, the weight of the vertical loads with respect to the horizontal ones ranged between 45% and 64%. Ordinary and weighted least square methods have been used to determine the hydrodynamic coefficients as a function of the experimental values of $u$, $a_H$, and $a_V$, and of $F_H$ and $F_V$. The WLS1 method has proved to be the best approach for the minimization of the difference between the force peaks and the related phase shifts by comparing experimental and calculated wave loads. A good agreement has been found for the positive horizontal and vertical forces which are fundamental for the stability analyses of the barrier. The analysis of the force coefficients has underlined a decreasing trend for $C_D$, $C_{MH}$, and $C_L$ vs. $A/d$, while $C_{MV}$ has highlighted an initial increasing trend and successive slight decreasing trend vs. $A/d$. For engineering aims, the hydrodynamic coefficients have been fitted by linear and non-linear laws by the knowledge of the wave non-linearity. The application of Morison and transverse formulas has paid attention to the single force components, highlighting an inertia-dominated regime in the wave direction and a lift-dominated regime in the transverse direction.

On the basis of experimental activities conducted by the Authors with $a/d = 0.7$ and 0.9, further investigations will be addressed to calibrate semi-empirical formulas, taking into account the breaking effects for the calculation of the solitary wave loads at submerged barriers having low submergence.

**Author Contributions:** F.A., G.T., L.G. and P.F. conceived and designed the experiments; G.T. performed the experiments; F.A., G.T. and L.G. analyzed the data; F.A. wrote the paper; and F.A. and P.F. supervised the research. All authors have read and agreed to the published version of the manuscript.

**Funding:** This research received no external funding.

**Institutional Review Board Statement:** Not applicable.

**Informed Consent Statement:** Informed consent was obtained from all subjects involved in the study.

**Data Availability Statement:** The data presented in this study are available on the request from the corresponding author.

**Acknowledgments:** This research was supported by the *PON*03*PE*000121 project called "Marine Energy Laboratory" and the project SILA (Integrated System of Laboratories for the Environment)—Research and Competitiveness 2007–2013 for Convergence Regions—which was funded by the Italian Ministry of Education, University and Research (MIUR). We are indebted to the laboratory technicians of University of Calabria Fabio De Napoli, Franco Leone and Salvatore Straticó (GMI Laboratory of the Department of Civil Engineering) for the support in building the physical model for the experimental tests.

**Conflicts of Interest:** The authors declare no conflict of interest.

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
