# Peer review of "Determination of Force Coefficients for a Submerged Rigid Breakwater under the Action of Solitary Waves"

_water, doi:10.3390/w13030315_

Round 1

Reviewer 1 Report

This paper presents an analysis related to the evaluation of Morison and transverse force coefficients in the case of a submerged square barrier subject to the action of solitary waves. The topic is of interest to researchers and engineers in the areas of marine and hydrokinetic energy system and this paper is well organized. The reviewer's main concern about this paper is its significance and originality. The reviewer would suggest the authors to explain the motivation of this research, why is this research needed and what values the research outcomes will bring to the sustainable energy research community and industry? Also, the originality of this research has to be discussed, the authors used existing mathematical models to calculate hydrodynamic forces acting on the submerged rigid breakwater but the reviewer is not clear about any new knowledge and novel methods developed from the present study.

What is the size of the experimental set-up shown in Fig. 1? What types of wave can it generate? How were the wave flume and other auxiliary facilities designed and built? This device is similar to a wave generation system available at the University and Louisiana at Lafayette, please refer to:

i) Y.-C. Liu, G. Cavalier, et al., "Design and construction of a wave generation system to model ocean conditions in the Gulf of Mexico", International Journal of Energy and Technology, 4(31), 2012, 1-7.

ii) Y.-C. Liu, T.A. Kozman, et al., "Establishment of a wave energy and technology lab to promote the experimental study of ocean and wave energy", IMECE 2013-62968, ASME 2013 International Mechanical Engineering Congress & Exposition, San Diego, CA, USA, Nov. 13-21, 2013.

A number of typos and grammatical errors were detected when reviewing this paper, an incomplete list of those small errors and the reviewer's suggestion are listed as follows:

  1. Ln 168 and Ln 240, “proportionally” should be “proportional”.
  2. Ln 191-193, please rewrite the sentence “its overall shape resembles….” to make it clearer.
  3. Ln 209, “coefficient” should be “coefficients”.
  4. Ln 234, does “OLS” mean “ordinary least square”? please spell it out.
  5. Ln 248, “choice” should be “choose”
  6. Ln 265, “view” should be “viewed”.
  7. Ln 267, please correct grammatical error in “A/d about equal to 0.22”
  8. Ln 270 to 273, please break this sentence into 2 to 3 short sentences to make it easy to understand.
  9. Ln 280, should be “IS very close to…”
  10. Ln 333, “catch” should be “caught”.

Reviewer 2 Report

The present study is to determine the force coefficients for a submerged rigid breakwater under the action of solitary waves using Laboratory tests. The research results of the authors are impressive. I only have a comment as follows:

The uncertainties of the present study are better to be discussed.

Reviewer 3 Report

This paper describes findings from 18 2D laboratory tests to calibrate the Morison and transverse semi-empirical models to determine the horizontal and vertical hydrodynamic forces from a submerged rigid breakwater under the action of solitary waves. Results are clearly presented with solid proof. I kind of enjoy reading this manuscript. I would like to suggest a minor review. Details are listed below.

Abstract, instead of listing what they did in this manuscript, I would suggest briefly summarize the conclusions, highlighting the most creative and exciting findings.

Line 39: need to justify why ratio of a/d was set to be 0.5.

Line 39-45: these belongs to model-setup, may be moved to session 3.

Line 43: is the “A” in ”A/d” here the same with the one inte line 30 (a/d)? If not, please describe the meaning of A.

Line 155: occur -à occurred

Line 181: please add a figure which can be referred to in terms of the comparison between theoretical equation and the modeled results.

Line 184-186: please add some explanation of the higher positive peak and lower negative through of the horizontal force.

Reviewer 4 Report

In this paper, the authors determined the coefficients of Morison equations from 18 experiments under solitary wave forcing. Wave gauges and pressure transducers are equipped to measure the surface elevation and pressure. They end up with the empirical formula for the coefficients with respect to nonlinearity (A/d). Overall, the paper is written in a clear way. I do not have any technical issues as it looks sound. The only suggestion is that if the author can comment on the present results (the coefficients in Fig. 10) compared to the periodic forcing. Also, how sensitive is the number of pressure gauges used in the analysis?   
